# Bubble Evolution under the Action of Polycarboxylate and Air-Entraining Agent and Its Effects on Concrete Properties—A Review

**DOI:** 10.3390/ma15207053

**Published:** 2022-10-11

**Authors:** Shuncheng Xiang, Yansheng Tan, Yingli Gao, Zhen Jiang, Bin Liu, Wei Zeng

**Affiliations:** 1School of Transportation Engineering, Changsha University of Science & Technology, Changsha 410082, China; 2CSCEC Western Construction Hunan Co., Ltd., Changsha 410082, China

**Keywords:** concrete bubble, bubble evolution, concrete properties, air-entraining agent, polycarboxylate acid structure

## Abstract

In order to improve the performance of concrete, it is of great significance to have a better understanding the mechanism and main influencing factors of concrete bubble evolution under the action of polycarboxylate and air-entraining agents. In the present review, with respect to the generation, growth, stability, and rupture of concrete bubbles under the action of polycarboxylate and air-entraining agents, this paper discusses the influence of bubble characteristics on concrete performance and studies bubble regulation by air-entraining agents and polycarboxylate (PCE) superplasticizer. The results show that the acid-to-ether ratio, sulfonic acid group, ester group, and the type of air entraining agent of the polycarboxylate acid structure have a significant impact on the bubbles. The bubble size, specific area, spacing factor, and bubble content have a significant impact on the rheological properties and related mechanical properties of fresh concrete and also affect the appearance quality of concrete. The problems with the experimental methods and theoretical models of concrete bubble research were analyzed, and future research ideas were put forward.

## 1. Introduction

For concrete, as a multiphase and porous material, pore structure is an important part of its microstructure which affects multiple of its properties. By adding an appropriate number of bubbles into concrete through admixtures, it is possible to increase the fresh concrete workability and effectively enhance its frost resistance and durability, but too many bubbles will also significantly reduce the concrete strength. Hence, it is particularly important to study the evolution mechanism and influencing factors of concrete bubbles. As a third-generation high-performance water reducer, PCE superplasticizer has various molecular structures. PCE with a special structure can be used as an air-entraining water reducer. Factors such as molecular mass, main chain density, and side-chain length significantly affect its air-entraining properties and the structure characteristics of the generated bubbles [1]. At the same time, air-entraining agents of different types create different effects on the characteristics of concrete bubbles, resulting in different concrete properties [2]. Scholars at home and abroad have conducted thorough research on the effects of concrete bubbles under the action of PCE, as well as the influence of co-doping defoamers and air-entraining agents in different proportions on concrete bubbles. However, the whole process mechanism of bubble generation and development is barely studied. In addition, the research mechanism of foam stability is not fully applicable to bubble stability in fresh concrete [3], and there is no relevant research or systematic reports on the mechanism of bubble stability in concrete.

The traditional experiment process consumes a lot of time and energy. Nowadays, given the rapidly developing network and computers, developed computer technology can be used to establish appropriate models which provide an intuitive and low-carbon experiment approach to simulate the progress of related experiments and finally reveal the evolution and regulation mechanism of concrete bubbles under the action of PCE. Focusing on the concrete bubble research by domestic and foreign scholars, this paper summarizes the effects of air-entraining agents and the PCE superplasticizer structure on bubbles, as well as the effect of bubbles on concrete properties. Moreover, it anticipated experimental methods and mathematical models for future research of bubble mechanisms. Our research can be reflected in Figure 1.

## 2. Research Progress in the Mechanism of the Whole Bubble Development Process

### 2.1. Types of Concrete Bubbles

There are generally two or more flow states involving mixtures of different phases in nature; the most common is the gas–liquid two-phase flow state. In this process, bubbles play an important role [4]. Bubbles in concrete can be divided into water bubbles, air bubbles, etc. [5]. Blisters usually appear when the cement amount is small, a phenomenon caused by water cavities formed by the later evaporation of water wrapped in the flocculation structure. Bubbles as inherent bubbles in concrete mixtures are mainly formed due to oily release agents, unreasonably graded aggregates, and stirring. Normally, the bubbles introduced by the PCE superplasticizer are irregular and unstable, which are more likely to lead to coalescence during concrete vibration and transportation and are harmful bubbles. The bubbles introduced by the air-entraining agent have a dimension of 20–200 μm, which are relatively stable and have firm liquid film to effectively prevent the aggregation, and belong to beneficial bubbles [6].

### 2.2. The Generation and Growth of Concrete Bubbles

According to the volume change of bubbles, the bubble life cycle has four stages: generation, growth, stability, and rupture. The generation of concrete bubbles involves both macroscopic and microscopic factors. Macroscopically, the artificially or mechanically created vortex swirls into the air to form a large number of bubbles with an uneven volume, and a small number of bubbles comes from the air dissolved in the water [7]. Microscopically, the air-entraining and dispersion effects of admixtures can introduce bubbles and promote their generation. The surface tension and diffusion properties of the air-entraining agent in the gas–liquid boundary play an important role in forming its air-entraining ability [8]. The various functional groups in the structure of the PCE superplasticizer greatly affect its air-entraining performance [9]. The Ca^2+^ concentration in the cement slurry increased continuously until reaching the maximum value near the end of the induction period. The air solubility in the pore solution lowered during the induction period, with air released and bubbles formed [10]. The relevant mechanism of action is shown in Figure 1 below.

There are two commonly used nucleation mechanisms in bubble evolution: 1. Classical nucleation theory, including two cases of homogeneous nucleation and heterogeneous nucleation; this theory is limited to static fluids in the scope of the application [11]. However, when bubble nucleation occurs, not all nucleated bubbles can continue to grow. Only when the bubble nucleation radius exceeds the critical value, the bubble is stable and can continue to grow. 2. The nucleation theory of shear flow field. Some scholars believe that the shear stress in the flow field effectively promotes the bubble nucleation [12], and the concrete during the mixing process is subjected to shear force, thus belonging to the scope of shear flow field. There is no unified conclusion as to whether the above bubble nucleation theory can be applied to complex concrete flow systems. In addition, the “cellular model” is a mathematical model describing bubble growth, and similar computer models can be established to study the bubble generation mechanism in concrete [13].

Essentially, bubble growth in concrete is when the gas surrounding the bubble diffuses into the bubble interior through the bubble film, generating a pressure difference between the interior and exterior. Scholars generally believe that the pressure gradient between bubbles leads to gas diffusion. Small bubbles have a high gas pressure, while large bubbles have a low gas pressure. Macroscopically, small bubbles converge to large bubbles and gradually become smaller until their disappearance, while large bubbles gradually become larger until they burst. The schematic diagram of gas diffusion is shown in Figure 2 [14].

### 2.3. Movement of Concrete Bubbles

In their process of formation, concrete bubbles will be subjected to upward buoyancy, downward surface tension, bubble inertial forces and drag forces, etc. Bubbles have vertical and horizontal movements. In vertical movements, the rise of bubbles is negatively correlated with the liquid phase viscosity, a process affected by buoyancy, viscous resistance, and self-gravity [3], as shown in Figure 3. In the rising process, the bubble force was analyzed according to the Navier–Stokes equation [15], and its rising speed was proportional to the buoyancy force and inversely proportional to the frictional force. However, the bubble did not perform linear motion in the rising stage, but swung upward along the central axis. Under greater viscous resistance, the trajectory presented greater linear tendency [16].

### 2.4. Stability and Rupture of Concrete Bubbles

Bubbles have unobvious boundaries between the growth and stabilization stages, and the volume does not change significantly in the stabilization stage. Many factors affect the stability of concrete bubbles including separation pressure and ionic strength, bubble liquid film strength [14], and the diffusion rate of gas through the liquid film.

Separation pressure and ionic strength. Separation pressure refers to the electrostatic repulsion between the two sides of the bubble liquid film, which consists of van der Waals forces and short-range repulsion [17,18]. Carey et al. [19] found through research that ionic active agents could increase the electrostatic repulsion under separation pressure. When the liquid film became thinner, bubble stability was mainly determined by the electrostatic repulsion between the ion adsorption layers on both sides of the liquid film [20]. When the cement slurry was hydrated, the Ca^2+^ concentration increased during the induction period and the bubble liquid film produced electrostatic repulsion [21] which hindered the coalescence of the bubbles, resulting in a smaller bubble dispersion radius, thus contributing to bubble stability [22].

Bubble liquid film strength. Film strength is often determined by the surface elasticity and surface viscosity of the adsorption film on the liquid film surface [23,24,25]. Surface elasticity refers to the resilience performance of the liquid film when it is locally stretched or thinned under the action of an external force. Special elasticity is a result of migration of film-forming materials [20]. Sett et al. [26] found through experiments that a liquid film with a high surface elasticity had a longer life, and vice versa. Higher surface elasticity means higher bubble stability. Surface viscosity refers to the viscosity within the surface monolayer of a liquid film [27,28] which is mainly derived from the interaction and hydration of active-agent molecules between hydrophilic groups in the surface monolayer. Under higher viscosity, the liquid film had a significantly smaller drainage rate, with bubble stability enhanced [20].

Diffusion rate of gas through the liquid film. The barrier to gas migration between the gas–liquid interface in fresh concrete, namely, the ability of gas to penetrate the liquid film, determines the bubble diffusion rate [29]. Gas diffusion occurs in the water-containing pores between surface-active molecules [30]. When the molecules are more densely arranged on the bubble film surface, gas has greater difficulty to penetrate the liquid film, leading to greater bubble stability [31].

Deformation is common during the concrete bubble movement, and bubble deformation is the premise of instability and rupture [32]. The water in the cement slurry flows over the surface of the rising bubbles, so the surfactant molecules are brought to the bottom of the bubbles creating a surface gradient between the top and bottom of the bubbles, which is the so-called Marangoni effect [33]. Studies have shown that the strength of the concrete bubble liquid film is the most important factor affecting the bubble stability, which carries the most research value. Greater film strength indicates better bubble stability [14]. The results of many scholars’ research regarding bubble stability are shown in Table 1 and Table 2 below.

The stability of concrete bubbles is dynamically balanced. Nonetheless, when the degree of bubble deformation causes the bubble film to rupture and lose its elastic properties, bubble will lose stability and rupture and finally exist in the interior and surface of the hardened concrete in the form of open or closed pores. There are three main reasons for the instability and rupturing of bubbles [40]: coalescence, liquid film drainage, and coarsening.

Coalescence. According to the classic model of bubble coalescence in aqueous solutions [41], bubble coalescence can be viewed as a process of liquid film drainage in which long side-chain PCE is used as surfactant, and its side-chain molecular structure will cause gas–liquid interface changes in the chemical and physical properties [42,43]. Since long side-chain PCE has a greater relative molecular mass than ordinary PCE, it is less likely to retain in the free solution of fresh concrete under the same conditions. As a result, its surface tension gets greater and the bubble liquid film gets thinner, resulting in an earlier bubble coalescence time [44] as shown in Figure 4. CT three-dimensional reconstruction technology was used to observe the evolution of local bubble morphology in the concrete mortar under ordinary PCE, ether long side-chain PCE, and ester long side-chain PCE, respectively [45]. It could be clearly observed that the adjacent bubbles coalesced in all the three groups of samples. In Figure 4, the bubbles corresponding to the common PCE coalesce at 30~60 min, while in Figure 4b,c, the bubbles corresponding to ultra-early strong PCE coalesce at 5~30 min.

Using computer CT three-dimensional reconstruction technology, the existence of the bubble coalescence phenomenon in fresh concrete was intuitively proved. Under the action of long side-chain PCE, the small pore-size bubbles in concrete decreased and the large pore-size bubbles increased. Moreover, long side-chain PCE caused earlier bubble coalescence time and earlier bubble rupture time.

Liquid film drainage. Drainage means liquid flow relative to the bubble [46]. Bubble drainage is accelerated by gravity and capillary pressure, which is also hindered by the viscous force [47]. The liquid discharge process can be free or forced [17,48]. The rapid discharge of liquid between bubbles makes the bubble film thinner. When the bubbles were just formed, the liquid film was thicker, which drained under the action of gravity and gradually became thinner. When the thickness of the bubble liquid film was reduced to a certain extent, the liquid film drainage was dominated by Plateau drainage [49]. When thinning reached a certain extent, the bubble burst [1].

Coarsening. Coarsening refers to the process in which bubble size increases through gas diffusion between adjacent tiny bubbles [50]. After coarsening, the bubble system had a decreased total surface area and more liquid was withdrawn from the bubble structure. When small bubbles broke into large bubbles more frequently, the liquid discharge rate increased under a higher coarsening rate, and the bubble film discharge rate affected the bubble film strength, which in turn resulted in bubble instability and rupture [25]. The total energy of the entire concrete bubble system decreased with increasing bubble size [51]. During the coarsening process, the bubble size increased and the number decreased. As shown in Figure 5 below, gas dissolves from surrounding small bubbles into greater bubbles, with smaller bubbles shrinking and greater bubbles expanding over time.

### 2.5. Research Progress in Computer Simulation

At present, most research on bubble behavior stays in the stage of clear water solutions. Appropriate mesh size and bubble parameters are important guarantees for reliable and effective simulation results, which are also difficult problems in simulation calculations. Nguyen et al. [52] studied validation experiments of bubble collapse and 3D simulations of bubble collapse near inclined walls. Bubble collapse was simulated under the hypothesis of five meshes and axisymmetric flow, with different numbers of units set for the initial bubble radius. Figure 6 shows a comparison of the bubble evolution during the numerically simulated collapse in the experimental image and the fine grid. Numerical results show that, by geometric interface reconstruction of VOF-PLIC, the sharp interface and collapse behavior of bubbles can be simulated, which is consistent with the experimental results. Figure 7 shows a visual comparison of the simulated and experimental images of the bubble growth process, and the numerical simulation results are in good agreement with the experimental results. In the later stages of bubble growth, the bubble began to take on an elliptical shape, and the upward-moving free surface interacted with the inclined walls, creating splashes at the corners between the free-surface walls and the dome. Using the VOF (Volume of Fluids) and GMO (General Moving Object Technique) models, Fantous et al. [53] successfully simulated the dynamic interaction between three different phases of cement paste, aggregates, and bubbles by the CFD (Computational Fluid Dynamics) method. Numerical simulations showed that both the diameter and the initial vertical position of the bubbles significantly affected the void system stability in both static and dynamic shear modes. Therefore, greater bubbles located in the top layer of the cement paste were more unstable than smaller ones initially placed in the bottom layer of the cement paste. At the same time, it was proved that the displacement and deformation characteristics of bubbles are more likely to be controlled by the shear mode and the shear source distance.

Some scholars carried out three-dimensional numerical simulations of the interactions between parallel bubbles under the VOF model [54]. A gas–liquid bubble column made of plexiglass was built to investigate experimentally the bubbling behavior including the bubble formation, growth, and rising motion. The experimental setups are shown in Figure 8. In order to eliminate the impact of light refraction on the visual measurements, the bubble column has a square cross section with a side length of 50 mm. Three types of gas distributors, including one-, two- and four-orifice distributors, are shown in Figure 8. The vertical distance between the gas distributor and the liquid distributor is 55 mm, and the chamber under the liquid distributor is 100 mm in height. Water in the storage tank was pumped by a centrifugal pump, regulated by a controlling valve, measured by a rotameter, distributed by the liquid distributor, and finally entered into the bubble column. Air was compressed by an air compressor, stabled in an air storage tank, regulated by a controlling valve, measured by a rotameter, distributed by the gas distributor, and ultimately passed into the bubble columns. Air and water co-currently flowed upward with the generated bubbles. Figure 9 below shows a comparison of the bubble behaviors in the case of a single well and a double well under the same operating conditions. The pictures at time t = 0.050 s, 0.225 s, and 0.375 s, respectively, showed the first separation of bubbles from the orifice, the first coalescence in the vertical direction, and the second coalescence in the vertical direction. For a period of time, there was no interaction between the double-pore and single-pore parallel-rising bubbles. Hence, bubbles with a large relative center spacing have negligible interactions. The two parallel bubbles did not merge, but the left and right bubbles had different rising speeds, presenting a non-linear rising path as shown with Figure 9. In order to detect different phenomena, the hole distance was adjusted to an appropriate distance (12.75 mm), as shown in Figure 10 below.The analysis shows that when the bubble diameter exceeds a certain value, the interaction is ignorable.

## 3. Factors Influencing Bubble Generation and Development

### 3.1. Influence of Different Types of Air-Entraining Agents on Concrete Bubbles

The behavior of concrete bubbles is affected by multiple factors such as different types of air-entraining agents and the PCE structure, raw material composition, construction technology, link temperature, ambient air pressure, etc. Air-entraining agents and PCE superplasticizer have the most significant effects. Using the image analysis method, Rath et al. [55] tested the bubble structure parameters under the action of different air-entraining agents and performed RGB image acquisition, image enhancement, image grayscale processing, binarization, object segmentation, and measurement analysis operations. The results showed that type of air-entraining agent significantly affected the correlation between the air void of fresh concrete and hardened concrete, as well as the average bubble diameter. As shown in Figure 11, the main components of air-entraining agents A and C are triterpene saponins and mixtures of alkyl ethoxyammonium sulfate and fatty alcohol, and the main components of air-entraining agents B and D are anionic and non-ionic mixtures such as sodium dodecyl sulfate, fatty alcohol, and polyether. The fresh concrete introduced with air-entraining agents A and C has a higher correlation with hardened concrete in terms of the air void, while the addition of other types of admixtures makes the correlation lower. It suggests that bubble content, stability, and bubble liquid film strength are greatly different when different types of air-entraining agents are introduced.

Under the same air void in the fresh concrete, the addition of different types of air-entraining agents also created obviously different effects on the average bubble diameter, as shown in Figure 12 and Figure 13. It suggests that it is inappropriate to characterize the state of concrete bubbles using only a certain index such as air void.

Atahan et al. [56] compared the influence of four different types of air-entraining agents on the pore distribution and bubble spacing coefficient of fresh concrete after bubble hardening. The bubble spacing coefficient is an important parameter for evaluating concrete frost resistance. Figure 14 shows that bubbles introduced with different types of air-entraining agents have varying sizes. Comparative analysis with Figure 15 found that the introduction of more tiny bubbles better helped to reduce the bubble spacing coefficient of concrete, thereby improving concrete frost resistance.

Qiao et al. [57] and Chen et al. [58] reported gemini surfactants as novel air-entraining agents for concrete. The gemini surfactants with different anionic groups, hydrophobic chains, and spacer groups were prepared and discussed. Chen et al. [59] also reported cationic oligomeric surfactants as novel air-entraining agents for concrete. The results indicate that, compared with traditional single-chained surfactants, the gemini and oligomeric surfactants have a higher foaming ability, foam stability, and air-entraining performance. Shan et al. [60] reported the synergism effects of nonionic and anionic surfactants for entraining stable air bubbles into fresh concretes. Qiao et al. [61] reported the effects of salts and adsorption on the performance of the air-entraining agents with different charge types in both solutions and cement mortars.

### 3.2. Influence of the PCE Structure on Concrete Bubbles

Most PCE graft copolymers contain carboxylic acid adsorption groups on the main chain, while the polyether side chains on the branched chains provide steric groups. Hydrophilic side-chain polyether groups have a strong air-entraining performance. When PCE was added to the concrete, the surface tension of the mixing water decreased, and a large number of unstable bubbles were introduced during the mixing process. Low surface energy facilitated the formation of harmful large bubbles. The bubbles entrained by different molecular PCEs had varying stabilities. For PCEs with carboxyl groups and ester groups, the bubbles easily ruptured, but this was not the case for PCEs with ester groups. It suggests that bubble regulation is closely related to the PCE structure.

Acid-to-ether ratio. He Yan [62] synthesized a series of PCEs in different acid-to-ether ratios by aqueous solution radical polymerization. Greater acid-ether ratios increased its adsorption capacity on the cement particle surface, which reduced the cement particle–water and solid–liquid interfacial energy and significantly increased the air entraining performance of PCE, thus significantly improving the pore-size distribution of bubbles. With the increase in the acid-ether ratio, cement mortar had a significantly increased air void. Under an acid-ether ratio of 4:1, the air void in the cement mortar reached the maximum value. Therefore, PCE could significantly improve the pore-size distribution of hardened mortar, introduce excessive medium and small bubbles, and reduce the proportion of large pore-size bubbles so that the pore-size distribution of the bubbles was finer. At the same time, with the increase in the acid-ether ratio in the main PCE chain, the proportion of introduced small pores decreased, while the proportion of macropores increased, as shown in Figure 16.

Sulfonic acid group. When a small number of sulfonic acid groups was used to replace the carboxyl groups in the main PCE chain, the fresh mortar with sulfonic acid group-containing PCE had a significantly increased or decreased air void of 5–6%, as shown in Figure 17 below [63]. As the sulfonic acid group gradually increased, the air-entrainment amount in the cement mortar gradually decreased because more sulfonic acid groups would lead to an increased concentration of PCE adsorbed on the cement particle surface. With it, the concentration of the residual PCE in the pore solution decreased, thereby reducing the surface tension of the gas–liquid interface so that the air void introduced into the cement mortar decreased. Therefore, PCEs with sulfonic acid groups have low air-entraining performance.

Ester group. The ester group is a neutral group that reduces the ionic density and adsorption capacity in the PCE molecule. As the content of ester groups grafted onto the side chain of PCE increased, the concentration of PCE adsorbed on the cement particle surface decreased, while the concentration of PCE left in the pore solution increased, resulting in significantly decreased surface tension at the gas–liquid interface, thereby increasing the air void entrained in the fresh cement mortar. The ester group-containing PCE, as shown in Figure 18 below, has significantly improved air-entraining performance compared with ordinary PCE, with the air void of fresh cement mortar up to 6–10%.

In recent years, some scholars have synthesized low air-entraining PCEs by adjusting the molecular structure or molecular weight of PCE, such as by introducing the antifoaming functional group, PPGA400 (poly (propylene glycol) diacrylate), into the molecular structure [64] using the aqueous solution radical polymerization method to adjust the synthesis of PCE with different acid-to-ether ratios [62] or by performing graft random copolymerization of EO/PO (ethylene oxide/propylene oxide copolymer) polyether side chains on the main chain [65], etc. The above experimental methods have respective advantages and disadvantages. By molecular shearing, it is possible to reduce the gas content, effectively inhibiting the bubble generation, but the synthesis process is relatively complicated. After the grafted ester group enters the side chain, PCE has a lower surface tension, which can significantly improve the air-entraining ability and change large bubbles into small bubbles. PCE with sulfonic acid groups inserted in the main chain cannot optimize the pore structure of concrete, with a large number of macropores still present in the hardened cement mortar [63]. Therefore, how to use the above experimental methods to optimize the PCE molecular structure is still worthy of further study.

## 4. Influence of the Structure of Concrete Bubbles on the Appearance

Bubble parameter is an important condition for establishing bubble structure as an important index affecting the concrete appearance. There are generally three parameters to characterize the bubble structure: gas content, average bubble diameter, and bubble spacing coefficient. Bubble structure is commonly controlled by the control of raw materials, adjustment of the mixing ratio, and composite mixing of chemical admixtures, among which the use of admixtures is the most efficient [66]. Under the basically consistent working performance, the effects of two traditional air void control methods, direct defoaming and first elimination and then introduction, on the apparent bubbles of hardened concrete were studied. The research results of Puthipad et al. [67] are shown in Figure 19 and Figure 20. Using the control method of “first elimination and then introduction”, the ratio of bubble area to the analysis area and the number of bubbles above 1 mm were greatly reduced, which played a significant role in improving the concrete surface appearance. The composite mixing of defoamer and PCE superplasticizer greatly affected the bubble spacing coefficient and bubble pore-size distribution of the concrete [68,69], among which the foaming and foam stabilization properties of the air-entraining agents played the most important role in affecting the concrete appearance [70]. A comprehensive comparison of Figure 21 reveals that the combined use of chemical admixtures can achieve a dynamic balance between improving and reducing workability, optimize the bubble structure, and enhance the bubble stability, thereby improving the concrete appearance quality [32].

## 5. Effect of Bubbles on Concrete Properties

### 5.1. Influence of Bubble Size, Specific Area, and Spacing Factor on the Rheological Properties of Concrete

Both durability and construction quality of concrete are affected by its rheological properties [71]. Guo et al. [72] reported the characteristic analysis of air bubbles on the rheological properties of fresh cement mortars and indicated that the bubble deformation decreases yield stress and plastic viscosity of the mortars. Using the NXS-11B rotational viscometer, Zhang et al. [73] measured the mortar plastic viscosity and yield stress and analyzed the influence mechanism of bubble characteristic parameters on rheological properties. The results showed that, with concrete specimens D-2 and F-1 as comparison samples as shown in Figure 22 and Figure 23, F-1 contained more bubbles when it was 10–600 μm, with a large number of tiny bubbles playing a lubricating role, which reduced the friction between the concrete components, thereby reducing the mortar yield stress and plastic viscosity. The tiny bubbles can separate the cement particles and decrease the degree of mutual colloidal interactions, which in turn breaks the cross-linked network and improves the rheological properties of concrete [74]. Figure 24 shows that the bubbles in F-1 have a smaller spacing factor and a larger specific surface area. At this time, a strong interaction occurred between the electrostatic repulsion generated by the same charge of the liquid films on the bubble surface so that the bubbles were uniformly dispersed in the concrete, which reduced the probability of small bubbles coalescing into large bubbles.

### 5.2. Effect of Bubble Content on Concrete Compressive Strength, Splitting Tensile Strength, and Drying Shrinkage

Khoshroo et al. [75] found through experiments that under a higher bubble content, the concrete compressive strength was lower, but the flexural strength did not change significantly, as shown in Figure 25 and Figure 26. Pressure was applied to the concrete test block to increase its air void and reduce its bearing area, thereby affecting the bearing capacity and decreasing the concrete compressive strength. The concrete flexural strength was mainly affected by internal pores and cracks [76], and the existence of pores and cracks resulted in an extremely fragile interface transition zone between the aggregate and the cement paste, but tiny bubbles could reduce the number of cracks in the concrete, thereby maintaining or enhancing the concrete flexural strength.

When fresh concrete contains a large number of uniform and tiny bubbles, energy can be better transferred under vibration and compaction. By analyzing the experimental data, Haddad et al. [77] found that the appropriate addition of chemical admixtures would introduce a large number of stable small bubbles, with the loss rate of compressive strength greater than that of the splitting strength, indicating increased concrete toughness. As shown in Table 3, the control group B with more air-entraining agents exhibits better toughness.

A large number of evenly distributed tiny bubbles can relieve the compressive stress on the solid phase particles due to an increased surface tension, which helps to reduce the drying shrinkage and deformation of concrete [78]. Maghfouri et al. [79] found through experiments that as the bubble content increased, concrete was less likely to undergo drying shrinkage and deformation. Although this trend is not obvious in the early stages of dehydration and drying (before 7 d), the later change trend is significant, as shown in Figure 27.

### 5.3. Effect of the Bubble Spacing Coefficient on Concrete Frost Resistance

Durability has become a major issue for concrete materials and structures in cold regions [78,80]. A low water-binder ratio can result in a more compact concrete structure [81,82], and the introduction of fine and discontinuous bubbles will increase concrete frost resistance. At present, some progress has been made in the research of chemical admixture effects on concrete frost resistance [83]. Bowers et al. [84,85,86,87] proposed two classical theories to illustrate the concrete frost resistance. According to the water pressure theory, the liquid in the void can freeze without damaging the concrete structure. The water in pores freezes first, forcing the liquid to move to other parts of the concrete structure. No unified research conclusion has been reached on the mechanism of concrete frost resistance. Scholars attribute the damage to a combination of factors including hydraulic buildup that forces the water away from the freezing point, osmotic pressure gradients that force the water to move toward the freezing point, vapor pressure potential, etc. [88]. Yang [89] found that the bubble spacing coefficient has a significant impact on the frost resistance durability of concrete, and when the spacing factor is >300 μm, the freezing and thawing resistance of concrete is relatively poor. Chang-Seon, S.A. [90] determined the void system of the freeze–thaw resistance of binary (ordinary Portland cement containing fly ash or silica fume) and ternary (ordinary portland cement + silica fume (5% fixed) + fly ash (changed to 45%)) mixed concretes. When the durability coefficient was 60%, the critical bubble spacing coefficient of the binary and ternary mixed concretes were 198.2 μm and 305.2 μm. Wang [91] believed that the improvement of the freeze–thaw durability of concrete depends on the bubble spacing coefficient of 0.22 μm–0.28 μm. Ley [92] observed that concrete with a SAM (sequential air method) number of 0.32 showed a durability coefficient between 60% and 80%, with a coincidence rate of 88%. In 68% of the survey data, 200 μm is related to a durability coefficient of 70%. In addition, scholars suggested that the number of SAM should be 0.22 to ensure that the concrete has good performance in the rapid freeze–thaw test.

Shon [90] found through experiments that the bubble spacing coefficient of concrete significantly affected the concrete frost resistance. When the bubble spacing coefficient was below 290 μm, the concrete frost resistance grade could reach F300. If the bubble spacing coefficient exceeded 300 μm, the concrete frost resistance grade was below F200. As shown in Figure 28, the bubble spacing coefficient is negatively correlated with the concrete frost resistance.

## 6. Conclusions and Expectations

To sum up, with the progress of cement hydration in fresh concrete, the physical and chemical properties of concrete experienced constant variations, with its bubble behavior greatly different from that in aqueous solution. The entire evolution process of concrete bubbles was simultaneously affected by multiple factors and then affected the rheological and mechanical properties of concrete. The current scholars’ research on the generation, growth, stability, and rupture of bubbles basically stayed in clean water solutions, and there is no literature on the in situ monitoring of concrete bubbles. Offline experiments need a lot of time and material costs, and it is more economical to use PFC software to simulate and study the impact of bubbles on the performance of concrete. In this work, the formation and rupture process of bubbles in concrete and the main influencing factors were described; the research progress on the effect of air bubbles on the performance of concrete was reviewed:The structural parameters of concrete bubbles greatly affected rheological and mechanical properties of concrete and the appearance quality after hardening. Chemical admixtures such as PCE and air-entraining agents had highly adjustable chemical structures, but the research on their structural design was not sufficient, so it is necessary to further improve the molecular structure and optimize the structural characteristics of the introduced concrete bubbles, thereby increasing the concrete durability and improving the pore quality after concrete hardening. We can also consider using PFC software to simulate the impact of the number and size of bubbles on the mechanical properties of concrete test blocks.Bubble stability was affected by bubble film strength. The viscosity, elasticity and permeability of bubble film could be used to characterize the bubble film strength. Research on the bubble film interface characteristics, liquid film drainage, bubble coalescence, etc., mostly target clean water; foam and cement mortar lack systematic quantification. In addition, the reported quantitative research on the stability of foam focused on the stability of foam. However, the mechanism of foam stability cannot be fully applied to the stability of foam in fresh concrete. There is no in situ observation or detection of the bubble behavior of fresh concrete, so it is applicable to study the bubble generation and development via the isotope trace method.The “cell model”, as well as the commonly used geometric interface reconstruction VOF-PLIC method, CT three-dimensional reconstruction technology, and the fluid volume VOF model, etc., are mathematical models established for foamed plastics, water, or two-phase systems. Concrete is a three-phase complex system of solid, liquid, and gas. Hence, it is necessary to set reasonable model hypotheses and parameters to qualitatively or semi-quantitatively describe the generation and development of bubbles.

## Data Availability

All the data used during the study are available from the corresponding author by request.

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
