# Peer review of "Bubble Evolution under the Action of Polycarboxylate and Air-Entraining Agent and Its Effects on Concrete Properties—A Review"

_materials, 2022, doi:10.3390/ma15207053_

Round 1

Reviewer 1 Report

This paper presents “Bubble evolution under the action of chemical admixture and its effects on concrete properties”. The topic is novel and the authors described the topic very well. However, in order to proceed for publication, following major comments should be incorporated.

The English writing of the manuscript needs improvement. Therefore, it could benefit greatly from professional editing to improve technical writing and English.

Please mention your study limits and suggest some future research topics.

In References, the sources are written in different styles. Please update the reference list. It is necessary to bring in accordance with the requirements of the magazine for the design of References. If possible, indicate DOI.

The literature can be expanded by studying some of these papers.

Prediction of compressive strength of rice husk ash concrete through different machine learning processes

Compressive strength prediction of rice husk ash using multi-physics genetic expression programming.

Please use some innovative keywords.

Please mention your study limits in the abstract.

The Conclusions should reflect what the practical application of the results obtained in this study is. In what climatic conditions should the recommendations of the authors be taken into account?

The authors should increase their discussion on previous related research and highlight how their study is providing a different approach or adding significantly to what has been done. The authors have to explain what is the new here in comparison with the previous studies. The novelty of the current work should be highlighted in the introduction. Please try to mention a problem that needs solving - in other words, the research question underlying your study clearer.

The title of the manuscript should be revised.

Some types of standards should be used to perform different experimental studies. Please provide details for the standards used in each study.

Section 4 should be discussed in detail.

The authors must redo the Abstract and bring it in compliance with the requirements of the journal. The scientific problem is poorly described (Background).

The scientific novelty is not indicated. I recommend shortening the Abstract to 200 words. Editors strongly encourage authors to use the following style of structured abstracts, but without headings: (1) Background: Place the question addressed in a broad context and highlight the purpose of the study; (2) Methods: Briefly describe the main methods or treatments applied; (3) Results: Summarize the article's main findings; and (4) Conclusions: Indicate the main conclusions or interpretations. The abstract should be an objective representation of the article.

It is advisable to add a flowchart at the beginning of the paper. Then the article would become more visual and structured.

The economic aspects are also required for sustainability in social aspect. It is suggested to authors to evaluate the cost-benefit study of this as a further investigation.

The conclusion should be an objective summary of the most important findings in response to the specific research question or hypothesis. A good conclusion states the principal topic, key arguments and counterpoint, and might suggest future research.

It is important to understand the methodological robustness of your study design and

report your findings accordingly. Please improve your conclusion section.

Reviewer 2 Report

Manuscript title: Bubble evolution under the action of chemical admixture and its effects on concrete properties

The paper is well-written, and the investigations reported in the paper are interesting and present a new topic related to the influence of bubble characteristics on concrete performance. All important aspects of the results are discussed. However, minor corrections should be considered before accepting the paper for publication, according to the following suggestions:

-          1- The research problem is missing in the abstract.

-         2-  Results un the abstract should be enhanced with focal findings.

-          3- Complementary discussion in comparison to literature is needed, to validate the study, show its novelty and gaps

Reviewer 3 Report

This is a review manuscript. Please add A Review in the title.

Please revise Sentence 55-57: Normally, the bubbles introduced by PCE superplasticizer are irregular in shape and uneven in size, which easily aggregate into large bubbles during the vibration and transportation, belonging to harmful bubbles.

There are no acknowledgement (reference/quote) for all figures.

Line 234-245 mentioned about a single well and a double well. However, it was not explained clearly about these two wells. Furthermore, there was no details explanation on Fig. 8a-f.

Please explain further details for Fig.9a-c

No explanation in the texts about Figure 10.

The authors have not included a Methods section explaining how the literature for review was selected, such as Content analysis; Grounded Theory; or Discourse Analysis.

Round 2

Reviewer 1 Report

Accept in present form